# Influenza Vaccination Coverage in Children: How Has COVID-19 Influenced It? A Review of Five Seasons (2018–2023) in Central Catalonia, Spain

**DOI:** 10.3390/vaccines12080925

**Published:** 2024-08-18

**Authors:** Sílvia Burgaya-Subirana, Mònica Balaguer, Queralt Miró Catalina, Laia Sola, Anna Ruiz-Comellas

**Affiliations:** 1Pediatrics Department, EAP Manlleu, Institut Català de la Salut, Gerència d’Atenció Primària i a la Comunitat de la Catalunya Central, C/Castellot, 17, 08560 Manlleu, Barcelona, Spain; sburgaya.cc.ics@gencat.cat; 2Faculty of Medicine, Universitat de Vic-Universitat Central de Catalunya, Cta/Roda, 70, 08500 Vic, Barcelona, Spain; aruiz.cc.ics@gencat.cat; 3Pediatric Intensive Care Unit, Hospital Sant Joan de Déu Barcelona, Passeig Sant Joan de Déu, 2, 08950 Esplugues de Llobregat, Barcelona, Spain; 4Research Department, Institut Català de la Salut, Gerència d’Atenció Primària i a la Comunitat de la Catalunya Central, C/Pica d’Estats, 13-15, 08272 Sant Fruitós de Bages, Barcelona, Spain; 5Medicine Department, EAP Sant Joan de Vilatorrada, Institut Català de la Salut, Gerència d’Atenció Primària i a la Comunitat de la Catalunya Central, Avinguda del Torrent del Canigó, 0, 08250 Sant Joan de Vilatorrada, Catalonia, Spain

**Keywords:** influenza vaccine, COVID-19, vaccination coverage, children

## Abstract

Influenza vaccination is the main method of preventing influenza. Vaccination is recommended for certain individuals with diseases that could cause complications in the case of flu infection. The objective of this retrospective observational study was to examine influenza vaccination coverage in patients with risk factors, to describe the characteristics of those vaccinated and to study the influence of COVID-19. The study population was children under 14 years old with risk factors between 2018/19 and 2022/23 in Central Catalonia, sourced through the electronic database of the Catalan Institute of Health. The association of influenza vaccination data with sociodemographic data and risk factors was performed by bivariate and multivariate analysis. A total of 13,137 children were studied. Of those, 4623 had received the influenza vaccine in at least one season. The average influenza vaccination rate was 28.8%. The statistically significant factors associated with vaccination were age and having certain risk factors: asthma, diabetes, haemoglobinopathies and clotting disorders. In all seasons, the immigrant population was vaccinated more than the native population *p* < 0.05, except for the COVID-19 season (2020/21), where no differences were observed. Of those vaccinated, 7.1% had been vaccinated for 5 consecutive years. Influenza vaccination coverage in the paediatric age group was low. Vaccination promotion measures must be implemented. COVID-19 meant an increase in vaccination of the native population to the same level as that of the immigrant population.

## 1. Introduction

Seasonal influenza is a disease with a major public health impact [1,2]. The World Health Organization (WHO) estimates that the annual influenza epidemic causes 1 billion infections, 3–5 million cases of severe disease and 290,000–650,000 deaths [1,3]. Children seem to be the most affected by this infection [4] and are the main transmitters of the virus [3]. In Spain, the highest cumulative incidence rate for influenza in the last two seasons (2021/22–2022/23) was observed in the 0–4 age group, followed by the 5–14 age group [5,6]. Influenza hospitalisations in the same period were highest in the elderly, followed by the 0–4 years age group [5,6].

Vaccination is the best way to prevent influenza [4]. In Catalonia (a region located in north-eastern Spain), up to the 2022/23 season, the Department of Health recommended influenza vaccination for all children between 6 months and 14 years of age with any condition at risk of suffering complications due to influenza virus infection [Table 1] as well as children from 6 months old who live with a person with any at-risk pathology [7].

Epidemiological studies suggest that 1 in 4 children have a chronic disease, with prevalence figures varying between 10 and 30% and, among them, asthma stands out as the main pathology with a prevalence of between 7 and 15% in Spain [8]. Obesity shows a generalised increasing trend in developed countries; in Europe it is a particularly serious problem in the countries of the South [8].

Despite the influenza vaccination policies implemented, influenza vaccination coverage in children with risk factors is low in our country [4,9,10,11,12,13,14,15]. A study conducted in the Community of Madrid during the 2018/19 season revealed influenza vaccination coverage of 15.6% [4], and another study conducted in Catalonia during the 2011/12 season showed vaccination coverage of 23.9% [9].

On the other hand, the COVID-19 pandemic, since its onset in December 2019, led to a reduction in routine childhood vaccination coverage worldwide [16,17,18,19,20]. The arrival of the pandemic caused a disruption in health services around the world, and Spain was no exception. Containment measures, the closure of health centres and the suspension of essential activities hindered access to health services, including vaccination programs, especially for the most vulnerable groups such as children under 2 years of age. In addition, fear of SARS-CoV-2 infection and uncertainty led to the increased reluctance of some parents to go to health centres to vaccinate their children with routine vaccines [18].

Childhood influenza vaccination coverage during the COVID-19 pandemic varied by country [21,22,23,24,25,26,27,28,29,30]. In the United States, for example, most studies report a decrease in influenza vaccination coverage [21,22,26,27]. As for the rest of the Americas, most countries also experienced a decrease in the percentage of influenza vaccination during 2020 [23]. Only Chile, Colombia and Peru experienced an increase in influenza vaccination coverage in children during the COVID-19 pandemic. South Korea, China and Italy also reported an increase in childhood influenza vaccination coverage [24,28,29,30]. As for Spain, no publications or data on the subject have been found.

The main objectives of this study were to describe childhood influenza vaccination coverage in Central Catalonia (Spain), to analyse the variables associated with vaccination, to study adherence to vaccination in children with risk factors in the last five seasons (2018–2023) and to evaluate the effect of the COVID-19 pandemic on childhood influenza vaccination.

## 2. Materials and Methods

### 2.1. Study Design and Participants

This was a retrospective observational study conducted in the healthcare district of Central Catalonia (Spain). 

The study included all children between 6 months and 14 years old with any risk factor [Table 1] for receiving the influenza vaccination and who were assigned to a primary care centre in Central Catalonia. This area provides healthcare to approximately 502,000 people. The number of children registered in this region is about 623,000. Data were collected for the last five seasons (2018/19 to 2022/23). These data were obtained from the electronic database of the Catalan health system (Catalan Institute of Health).

### 2.2. Variables

The variables analysed for each patient were as follows: age, sex, level of education of the mother and father, place of residence (rural or urban), place of origin (Spain or other), risk factors for receiving the influenza vaccination, number of risk factors for receiving the influenza vaccination, influenza vaccination status and adherence to influenza vaccination; influenza vaccination was the independent variable and the remaining variables were the dependent variables. 

To classify the place of residence in rural or urban area, we considered the number of people living in the village. A rural area is an area with fewer than 10,000 inhabitants. An urban area is an area with more than 10,000 inhabitants. 

The risk factors described in Table 1 were grouped into 10 categories: asthma, heart disease, coeliac disease, diabetes, haemoglobinopathies and clotting disorders, immunosuppression and neoplasms, neuromuscular and neurological diseases, chromosomopathies and metabolopathies, nephropathies and hepatopathies, obesity and miscellaneous.

Complete adherence was defined as having been vaccinated against influenza consecutively each year throughout the study period. For the calculation of this variable, children who could not have been vaccinated during 5 consecutive seasons due to their age were excluded.

To assess the effect of the COVID-19 pandemic on childhood influenza vaccination, the different variables studied and associated with vaccination were related to each vaccination season separately. If any variable was statistically significant or no longer statistically significant during the seasons coinciding with the COVID-19 outbreak, we considered whether COVID-19 had any effect on influenza vaccination in children.

### 2.3. Statistical Analysis

A descriptive and comparative analysis of the characteristics of vaccinated and unvaccinated children with risk factors from the 5 study seasons was performed and expressed as percentages, and a statistical significance of 95% was used. To calculate influenza vaccination coverage, children with risk factors who had been vaccinated against influenza were compared with all children who could have been vaccinated, i.e., all children aged 6 months to 14 years who had any condition putting them at risk and who were eligible for influenza vaccination. The average influenza vaccination coverage for the five seasons and for each season were calculated. 

To analyse the connection between the different variables and vaccination, a bivariate and multivariate analysis was performed. In the bivariate analysis, the Odds Ratio and its 95% confidence interval are shown and used, and in the multivariate analysis, the data were adjusted in a logistic regression. The selection criteria for the variables for the multivariate analysis for each season were as follows: that the data were statistically significant in the bivariate analysis in the last season, presence in more than 1% of the population and the clinical criteria of the researchers. 

Finally, for the calculation of vaccination adherence, the sum of the seasons that each child had received the influenza vaccine was calculated. If the sum was equal to 5, it was considered correct adherence. If the sum was less than 5, it was considered incorrect adherence. 

All differences were examined using confidence intervals and a confidence level of 95% was established. A *p* < 0.05 was considered statistically significant.

Statistical analyses were performed with program R version 4.2.1 (R Foundation for Statistical Computing). 

## 3. Results

### 3.1. Characteristics of Vaccinated Children with Risk Factors

Table 2 describes the characteristics of the 13,000 children aged 6 months to 14 years with risk factors for influenza vaccination. Of these, 4600 (35.2%) received the influenza vaccination in at least one season. Of those vaccinated, 2000 (42.1%) were girls. The ages of vaccinated and unvaccinated children were studied in a stratified manner, so the percentage of vaccinated children for each age group was as follows: 6 months to 2 years: 9.5%, 3–5 years: 18.4%, 6–10 years: 39%, 11–15 years: 33.1% Of those vaccinated, 50.8% lived in a rural area. Of the risk factors studied in vaccinated children, the following percentages were observed: asthma (58.8%); heart disease (4.3%); coeliac disease (5.87%); diabetes (4.3%); haemoglobinopathies and clotting disorders (1.15%); immunosuppression and neoplasms (1.95%); neuromuscular, neurological diseases, chromosomopathies and metabolopathies (1.02%); nephropathies and hepatopathies (0.72%); obesity (19.9%) and miscellaneous (2.11%). The country of origin was available for only 3530 vaccinated children, but the majority (84.8%) were Spanish. Most of those vaccinated (87.1%) had only one risk factor.

### 3.2. Influenza Vaccination Coverage in Children

The mean influenza vaccination coverage over the 5 years of the study was 28.8% [28.3, 29.3]. The season with the highest coverage was the 2020/21 season (coinciding with the COVID-19 pandemic): 37% 3 [5.8, 38.1] [Table 3].

### 3.3. Variables Associated with Influenza Vaccination

Table 4 shows the association of the different variables with influenza vaccination by season using bivariate and multivariate analysis. We observed that the variables age, number of risk factors and some of the risk factors such as asthma and diabetes were positively and statistically significantly associated with influenza vaccination in all seasons. Heart disease, coeliac disease and obesity were negatively associated with influenza vaccination in virtually all seasons, such that patients with these three conditions were statistically significantly less vaccinated against influenza.

In all seasons, it was observed that the younger the children, the more they were vaccinated.

The variables country of origin and haemoglobinopathies and clotting disorders are discussed in the following section because of the influence that COVID-19 had on influenza vaccination in these groups.

### 3.4. Influence of the COVID-19 Pandemic on Influenza Vaccination

In the 2020/21 and 2021/22 seasons, influenza vaccination coverage was significantly greater than in the 2018/19 season, coinciding with the COVID-19 pandemic. The coverage in these two seasons was 37% [35.8, 38.1] and 28.5% [27.5, 29.6], respectively [Table 3].

We observed that the variable “country of origin” is statistically significant in the pre-COVID-19 seasons, but that during the 2020/21 season (considered to be the first season of the COVID-19 pandemic) it is no longer associated with influenza vaccination in a statistically significant way. After this season, this variable returns to statistical significance, but only in the multivariate analysis. That is, before and after the COVID-19 pandemic, immigrant children were the most vaccinated against influenza, but during the pandemic, the vaccination of native children increased to equal that of immigrant children [Table 4].

In the 2021/22 and 2022/23 seasons, we observed that haemoglobinopathies and clotting disorders were beginning to be associated with influenza vaccination, OR 2.3 [1.3, 4.15] and ORO 3.5 [1.90, 6.4], respectively [Table 4].

### 3.5. Adherence to Influenza Vaccination

Of the 4623 children vaccinated over the five seasons studied, 49% of the children had been vaccinated in only one season, 21.6% in two seasons, 15.2% in three seasons, 7.2% in four seasons and 7.1% in all five seasons studied. Therefore, only 7.1% of vaccinated children had complete adherence to the influenza vaccination.

## 4. Discussion

The mean influenza vaccination coverage in children aged 6 months to 14 years with risk factors in Central Catalonia (Spain) in the last 5 years (2018–2023) was 28.8% [28.3, 29.3]. Although this result is slightly higher than other studies conducted in our country in previous seasons (23.9% in 2011/15 [9], 15.6% in 2018/19 [4] or 27.1% in 2009/10 coinciding with the influenza pandemic [11]), influenza vaccination coverage in children is low. One of the objectives defined by the Catalan Department of Health in the 2022/23 season was to achieve influenza vaccination coverage of 60% in people with risk factors [7]. For this specific period, coverage was 24.07% in Central Catalonia. In comparison with other European countries and the United States, the data on influenza vaccination coverage are very disparate. Coverage reported in Europe in a 2012 study ranged from 0.2% to 74% [31], and in the United States the 2022/23 season was 57.4% [32,33].

The highest vaccination coverage (36.96% and 28.55%) during the study period was in 2020/21 and 2021/2022, coinciding with the COVID-19 pandemic. We assume that this increase may be due to the similarity of the symptoms of influenza and COVID-19 and the fear that this provoked in families, in addition to the promotion of the influenza vaccination during the COVID-19 pandemic for people with risk factors. Some countries such as Italy, South Korea and China also experienced an increase in influenza vaccination during this period [24,28,29,30]. An Italian paper that studied influenza vaccination coverage in children between 2010/11 and 2020/21 showed that, in the 2020/21 season, influenza vaccination coverage in children increased 14.8% compared to previous seasons in the age group 2–4 years old (coverage 19%), followed by the group 5–8 years old and <2 years old (increase +10%, coverage of 13.1% and +6.4%, coverage of 9.2%, respectively) [28]. Another similar study conducted in South Korea showed that from 2019 to 2020 with the COVID-19 pandemic, influenza vaccination coverage of children aged 12–18 increased from 27.8% to 43.5% [24].

However, this increase was not observed in all countries. The Center for Disease Control and Prevention (CDC) in the United States observed a decrease in childhood influenza vaccination coverage of 5.1% compared to the previous season [34]. This decrease in childhood influenza vaccination coverage has been corroborated by other studies [21,22,23,26,27]. Fogel B et al. conducted an investigation in Southcentral Pennsylvania to determine whether early season rates of influenza vaccination changed in a season when there was a concurrent COVID-19 pandemic. They observed that early vaccination rates were lower in 2020 (29.7%) compared with 2018 and 2019 (34.2% and 33.3%, respectively) [21]. In the same line, Nogareda F et al. described in 2023 the seasonal influenza vaccination coverage in the Americas during 2019–2021. They reported that influenza vaccination coverage decreased by 9% for children [23]. To explain these differences between countries, it should be remembered that each country applies different influenza vaccination policies [7,31,34,35,36].

Regarding the variables associated with influenza vaccination, we observed that the younger the children are, the more they are vaccinated. This can be explained by the fact that younger children are more vulnerable and have more severe infections. Most hospitalisations in children have been recorded in children under 5 years old (especially children under 2 years old and, within this group, children under 6 months old) [5,6,37,38,39,40,41]. Paediatricians may place more emphasis on recommending influenza vaccination for younger children, for the reasons explained above. In addition, parents of children in these age groups with risk factors may be more fearful of the consequences of infection and more likely to vaccinate. Our study has shown that immigrant children are more likely to be vaccinated against influenza than native children. This result is similar to that observed in the study by González R et al. [9]. These differences can be explained by cultural reasons, due to the language barrier and the low educational level of the majority of immigrant parents (it has been observed that the rejection of the influenza vaccination is greater in parents with a high educational level [42]). During the COVID-19 pandemic, the differences in influenza vaccination between immigrant and native children disappeared, so native children were vaccinated at the same rate as immigrants. The main reason for this change may be fear of COVID-19. 

Asthma, diabetes, haemoglobinopathies and clotting disorders were the risk factors most associated with vaccination. Asthma is one of the most prevalent childhood diseases [43] with a high risk of complications and emergency department and hospital visits. Diabetes is a chronic disease that involves a lot of tests, which makes families experience it as a serious pathology. All this could explain why vaccination is more common in these groups. The increased rate of vaccination in patients with haemoglobinopathies and clotting disorders may be related to vaccination in immigrant children, since most of the children in this group had sickle cell disease, a disorder that more frequently affects certain minority ethnic groups from sub-Saharan Africa, India, Saudi Arabia and some countries in the Mediterranean area [44]. As for the number of risk factors, there is a positive relationship between vaccination and having more risk factors. This is logical since the more risk factors, the greater the risk of complications from the influenza virus. Heart disease, coeliac disease and obesity were negatively associated with vaccination in our study. In the case of coeliac disease and obesity it may be due to the lack of a sense of illness on the part of the family and the patient themselves. In the case of heart disease, poor vaccination is a cause for concern, and we found no explanation to justify it.

Finally, adherence to influenza vaccination in children in the different seasons is low in relation to the objectives proposed by the Catalan Department of Health. Most children were vaccinated in one season only (49%) and only 7.1% were vaccinated throughout the five seasons. Díaz-García R et al. found 65.9% adherence during the 2018/19 season and the following two seasons [4].

This study has limitations, the most important of which is that, since it is a descriptive study, it does not allow us to establish causal relationships between variables, but we can establish hypotheses. Another limitation is that we have only included children assigned to the ICS because we do not have information on children vaccinated by other health care providers or those covered by private insurance. In any case, the ICS provides health care for most of the children in our area, so the data are representative of our population.

As strong points, it should be noted that this is the first study of the effect of COVID-19 on childhood influenza vaccination coverage in Spain. This study was carried out in children with risk factors.

One line of research could be to observe whether immigrants of different races and from different countries have the same predisposition to vaccinate children against influenza.

## 5. Conclusions

Influenza vaccination coverage in children with risk factors in Central Catalonia is low, as is adherence to the influenza vaccination. It is necessary to change vaccination policies to improve coverage and adherence to vaccination in children. Some risk factors such as asthma, diabetes, and haemoglobinopathies and clotting disorders are associated with vaccination. Influenza vaccination is higher in immigrant children than in native children.

The COVID-19 pandemic positively influenced influenza vaccination in children, as vaccination coverage was higher and native children were vaccinated at the same rate as immigrant children.

## Figures and Tables

**Table 1 vaccines-12-00925-t001:** Risk factors that the Catalan Department of Health considered as indications for receiving the influenza vaccine in children older than 6 months up to the 2022/23 season.

Risk Factors for Influenza Vaccination in Children
- Chronic cardiovascular, neurological or respiratory diseases (including hypertension, asthma, bronchopulmonary dysplasia and cystic fibrosis). - Diabetes mellitus. - Morbid obesity: body mass index (BMI) ≥ 35 in adolescents and ≥ 3 standard deviations in children. - Chronic kidney disease and nephrotic syndrome. - Haemoglobinopathies and anaemia. - Haemophilia and other clotting disorders, chronic bleeding disorders, recipients of blood products and multiple transfusions. - Asplenia or severe splenic dysfunction. - Chronic liver disease. - Severe neuromuscular diseases. - Immunosuppression (including primary immunodeficiencies and those caused by HIV infection), drugs (including eculizumab treatment), or in transplant recipients and associated deficiencies. - Cancer and malignant blood diseases. - Cochlear implant or awaiting implant. - Cerebrospinal fluid fistula. - Coeliac disease. - Chronic inflammatory disease. - Disorders and diseases that entail cognitive dysfunction: Down’s syndrome, etc. - Children and adolescents receiving prolonged treatment with acetylsalicylic acid, due to the possibility of developing Reye’s syndrome after influenza. - Long-term institutionalised children. - Children between 6 months and 2 years old with a history of prematurity, born at less than 32 weeks gestation.

**Table 2 vaccines-12-00925-t002:** Demographic and clinical characteristics of vaccinated and unvaccinated infants with risk factors for receiving influenza vaccination between 6 months and 14 years.

	Study Population (N = 13,137)	
	Vaccinated (N = 4623)	Not Vaccinated (N = 8514)
Rurality (N = 13,137):		
Rural	2348 (50.8%)	4236 (49.8%)
Urban	2275 (49.2%)	4278 (50.2%)
Age (N = 13,137):		
6 months–2 years	439 (9.50%)	172 (2.02%)
3–5 years	851 (18.4%)	854 (10.0%)
6–10 years	1801 (39.0%)	3521 (41.4%)
11–15 years	1532 (33.1%)	3967 (46.6%)
Father education (N = 4936):		
No education	18 (1.49%)	90 (2.41%)
Primary education	295 (24.4%)	1024 (27.5%)
Secondary education	333 (27.6%)	1085 (29.1%)
Baccalaureate/Medium grade	184 (15.2%)	794 (21.3%)
University/high grade	353 (29.2%)	609 (16.3%)
Others	25 (2.07%)	126 (3.38%)
Mother education (N = 5041):		
No education	21 (1.70%)	110 (2.89%)
Primary education	202 (16.3%)	735 (19.3%)
Secondary education	317 (25.6%)	1046 (27.5%)
Baccalaureate/Medium grade	222 (17.9%)	801 (21.1%)
University/High grade	453 (36.6%)	1012 (26.6%)
Others	23 (1.86%)	99 (2.60%)
Risk factors (N = 13,137):		
Asthma	1533 (58.8%)	2826 (32.2%)
Heart disease	112 (4.30%)	256 (2.92%)
Coeliac disease	153 (5.87%)	452 (5.15%)
Diabetes	105 (4.03%)	44 (0.50%)
Haemoglobinopathies and clotting disorders	30 (1.15%)	31 (0.35%)
Immunosuppression and neoplasms	51 (1.95%)	73 (0.83%)
Neuromuscular, neurological diseases, chromosomopathies and metabolopathies	27 (1.02%)	42 (0.47%)
Miscellaneous	55 (2.11%)	65 (0.74%)
Nephropathies and hepatopathies	19 (0.72%)	28 (0.32%)
Obesity	518 (19.9%)	4947 (56.4%)
Gender (N = 13,137):		
Female	1948 (42.1%)	3632 (42.7%)
Male	2675 (57.9%)	4882 (57.3%)
Country of origin (N = 10,097):		
Spain	2993 (84.8%)	5805 (88.4%)
Others	537 (15.2%)	762 (11.6%)
Number of risk factors (N = 10,815):		
1	2005 (87.1%)	8265 (97.1%)
2	290 (12.6%)	247 (2.90%)
>2	6 (0.26%)	2 (0.02%)

**Table 3 vaccines-12-00925-t003:** Number of vaccinated children and seasonal influenza vaccination coverage.

	N (% Vaccinated/Susceptible to Vaccination)	IC 95%
2018–2019 (N = 6401)	1673 (26.1%)	[25.1, 27.2]
2019–2020 (N = 6871)	1916 (27.9%)	[26.8, 29]
2020–2021 (N = 7110)	2628 (37%)	[35.8, 38.1]
2021–2022 (N = 7031)	2008 (28.5%)	[27.5, 29.6]
2022–2023 (N = 6867)	1653 (24.1%)	[23.1, 25.1]
Global percentage (N = 34,280)	9878 (28.8%)	[28.3, 29.3]

**Table 4 vaccines-12-00925-t004:** Associated variables with seasonal influenza vaccination. Bivariate and multivariate analysis.

	2018–2019				2019–2020				2020–2021				2021–2022				2022–2023			
	BV		MV		BV		MV		BV		MV		BV		MV		BV		MV	
	OR (IC 95%)	*p*-Value	OR (IC 95%)	*p*-Value	OR (IC 95%)	*p*-Value	OR (IC 95%)	*p*-Value	OR (IC 95%)	*p*-Value	OR (IC 95%)	*p*-Value	OR (IC 95%)	*p*-Value	OR (IC 95%)	*p*-Value	OR (IC 95%)	*p*-Value	OR (IC 95%)	*p*-Value
**Age:**																				
**6 months–2 years**	**7.41 [5.6, 9.75]**	**<0.001**	**5.94 [3.92, 8.98]**	**<0.001**	**6.44 [4.94, 8.45]**	**<0.001**	**5.92 [3.95, 8.88]**	**<0.001**	**6.70 [5.16, 8.78]**	**<0.001**	**6.95 [4.69, 10.29]**	**<0.001**	**5.47 [4.25, 7.06]**	**<0.001**	**5.97 [4, 8.9]**	**<0.001**	**5.67 [4.38, 7.36]**	**<0.001**	**6.63 [4.43, 9.92]**	**<0.001**
**3–5 years**	**3.01 [2.54, 3.57]**	**<0.001**	**2.55 [1.97, 3.31]**	**<0.001**	**2.26 [1.91, 2.66]**	**<0.001**	**2.05 [1.6, 2.62]**	**<0.001**	**2.44 [2.10, 2.85]**	**<0.001**	**2.22 [1.73, 2.83]**	**<0.001**	**2.60 [2.22, 3.05]**	**<0.001**	**2.3 [1.8, 2.95]**	**<0.001**	**2.09 [1.76, 2.47]**	**<0.001**	**2.25 [1.73, 2.92]**	**<0.001**
**6–10 years**	**1.25 [1.09, 1.42]**	**<0.001**	**1.17 [0.96, 1.41]**	**0.113**	**1.27 [1.12, 1.43]**	**<0.001**	**1.16 [0.97, 1.38]**	**0.113**	**1.26 [1.13, 1.40]**	**<0.001**	**1.13 [0.95, 1.34]**	**0.155**	**1.33 [1.18, 1.50]**	**<0.001**	**1.3 [1.08, 1.56]**	**0.004**	**1.19 [1.05, 1.35]**	**0.008**	**1.07 [0.88, 1.31]**	0.505
**11–15 years**	**Ref.**	**Ref.**	**Ref.**	**Ref.**	**Ref.**	**Ref.**	**Ref.**	**Ref.**	**Ref.**	**Ref.**	**Ref.**	**Ref.**	**Ref.**	**Ref.**	**Ref.**	**Ref.**	**Ref.**	**Ref.**	**Ref.**	Ref.
**Country of origin:**																				
**Spain**	**0.55 [0.45, 0.68]**	**<0.001**	**0.47 [0.36, 0.63]**	**<0.001**	**0.70 [0.58, 0.84]**	**<0.001**	**0.56 [0.43, 0.73]**	**<0.001**	1.00 [0.84, 1.20]	0.984	0.81 [0.62, 1.05]	0.109	**0.99 [0.83, 1.18]**	**0.871**	**0.72 [0.56, 0.93]**	**0.012**	**0.92 [0.77, 1.10]**	**0.372**	**0.54 [0.43, 0.69]**	**<0.001**
**Others**	**Ref.**	**Ref.**	**Ref.**	**Ref.**	**Ref.**	**Ref.**	**Ref.**	**Ref**	Ref.	Ref.	Ref.	Ref.	**Ref.**	**Ref.**	**Ref.**	**Ref.**	**Ref.**	**Ref.**	**Ref.**	**Ref.**
**Number of risk factors:**																				
**1**	0.27 [0.06, 1.46]	0.119	**0.06 [0.01, 0.46]**	**0.006**	**0.21 [0.06, 0.77]**	**0.020**	**0.18 [0.03, 0.92]**	**0.040**	**0.08 [0.00, 0.59]**	**0.012**	**0.02 [0, 0.33]**	**0.005**	**0.21 [0.05, 0.92]**	**0.039**	**0.05 [0.01, 0.27]**	**<0.001**	**0.16 [0.04, 0.69]**	**0.017**	**0.03 [0.01, 0.15]**	**<0.001**
**2**	0.88 [0.18, 4.83]	0.872	0.62 [0.09, 4.28]	0.630	0.60 [0.16, 2.29]	0.447	1.14 [0.22, 5.83]	0.873	0.30 [0.01, 2.19]	0.255	0.28 [0.02, 3.66]	0.333	0.76 [0.17, 3.42]	0.710	0.46 [0.09, 2.39]	0.356	0.56 [0.13, 2.53]	0.439	0.26 [0.05, 1.38]	0.114
**>2**	Ref.	Ref.	Ref.	Ref.	Ref.	Ref.	Ref.	Ref.	Ref.	Ref.	Ref.	Ref.	Ref.	Ref.	Ref.	Ref.	Ref.	Ref.	Ref.	Ref.
**Risk factors ***																				
**Asthma**	**1.73 [1.55, 1.94]**	**<0.001**	**1.53 [1.09, 2.15]**	**0.014**	**1.43 [1.29, 1.60]**	**<0.001**	**1.45 [1.05, 2.01]**	**0.025**	**1.71 [1.55, 1.90]**	**<0.001**	**1.33 [0.98, 1.82]**	**0.069**	**1.92 [1.72, 2.14]**	**<0.001**	**1.47 [1.07, 2.01]**	**0.018**	**1.68 [1.49, 1.89]**	**<0.001**	**1.14 [0.82, 1.59]**	**0.421**
**Heart disease**	0.84 [0.61, 1.15]	0.292	0.89 [0.53, 1.48]	0.656	**0.71 [0.51, 0.97]**	**0.029**	0.67 [0.4, 1.11]	0.119	**0.66 [0.49, 0.88]**	**0.004**	**0.52 [0.32, 0.85]**	**0.008**	**0.61 [0.44, 0.83]**	**<0.001**	**0.52 [0.32, 0.86]**	**0.011**	**0.64 [0.46, 0.88]**	**0.005**	**0.51 [0.31, 0.84]**	**0.007**
**Coeliac disease**	**0.13 [0.08, 0.20]**	**<0.001**	**0.08 [0.04, 0.16]**	**<0.001**	**0.44 [0.33, 0.57]**	**<0.001**	**0.36 [0.22, 0.58]**	**<0.001**	**0.61 [0.49, 0.76]**	**<0.001**	**0.41 [0.27, 0.62]**	**<0.001**	**0.77 [0.61, 0.98]**	**0.031**	**0.51 [0.33, 0.79]**	**0.002**	**0.65 [0.49, 0.84]**	**0.001**	**0.39 [0.2,; 0.62]**	**<0.001**
**Diabetes**	**6.40 [3.95, 10.7]**	**<0.001**	**5.55 [2.86, 10.75]**	**<0.001**	**4.33 [2.78, 6.87]**	**<0.001**	**4.34 [2.39, 7.87]**	**<0.001**	**3.80 [2.38, 6.22]**	**<0.001**	**2.95 [1.59, 5.46]**	**<0.001**	**2.95 [1.96, 4.45]**	**<0.001**	**2.11 [1.21, 3.68]**	**0.008**	**2.30 [1.53, 3.45]**	**<0.001**	**1.32 [0.74, 2.33]**	**0.343**
**Haemoglobino-pathies and clotting disorders**	1.14 [0.47, 2.53]	0.752	-	-	1.37 [0.63, 2.80]	0.415	-	-	1.47 [0.77, 2.77]	0.241	-	-	**2.31 [1.28, 4.15]**	**0.006**	-	-	**3.49 [1.91, 6.38]**	**<0.001**	-	-
Immunosupres-sion and neoplasms	1.27 [0.74, 2.11]	0.373	-	-	1.14 [0.70, 1.81]	0.591	-	-	1.20 [0.77, 1.85]	0.419	-	-	0.74 [0.43, 1.20]	0.223	-	-	0.93 [0.54, 1.52]	0.771	-	-
Neuromuscular and neurological diseases, chromosomopa-thies and metaboloahties	0.64 [0.29, 1.27]	0.214	-	-	0.99 [0.47, 1.94]	0.977	-	-	0.78 [0.35, 1.62]	0.516	-	-	1.85 [0.91, 3.71]	0.090	-	-	2.02 [1.00, 3.94]	**0.049**	-	-
Miscellaneous	1.16 [0.70, 1.84]	0.557	-	-	1.30 [0.82, 2.02]	0.259	-	-	1.39 [0.90, 2.13]	0.142	-	-	1.06 [0.65, 1.69]	0.796	-	-	1.33 [0.80, 2.14]	0.266	-	-
Nephropathies and hepatopathies	1.76 [0.85, 3.50]	0.122	-	-	0.84 [0.37, 1.72]	0.649	-	-	1.52 [0.76, 3.01]	0.231	-	-	1.20 [0.54, 2.51]	0.639	-	-	1.22 [0.50, 2.67]	0.645	-	-
**Obesity**	**0.12 [0.10, 0.14]**	**<0.001**	**0.15 [0.11, 0.23]**	**<0.001**	**0.10 [0.09, 0.12]**	**0.000**	**0.16 [0.11, 0.23]**	**<0.001**	**0.07 [0.06, 0.08]**	**<0.001**	**0.1 [0.07, 0.14]**	**<0.001**	**0.08 [0.07, 0.09]**	**0.000**	**0.11 [0.08, 0.15]**	**<0.001**	**0.07 [0.06, 0.09]**	**0.000**	**0.09 [0.0, 0.13]**	**<0.001**

* Reference: to not have the risk factor. BV: Bivariate analysis. MV: Multivariate analysis.

## Data Availability

The data that support the findings of this study are available from the corresponding author upon request.

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
