# Peer review of "Influenza Vaccination Coverage in Children: How Has COVID-19 Influenced It? A Review of Five Seasons (2018–2023) in Central Catalonia, Spain"

_vaccines, 2024, doi:10.3390/vaccines12080925_

Round 1

Reviewer 1 Report

Comments and Suggestions for Authors

This study reviews coverage of vaccination against influenza among children under 14-ueal old with risk factors during COVID19 seasons. Five years (2018-2023 season were analyzed on 13,137 children from Central Catalonia and association of influenza vaccination with sociodemographic data and risk factors was performed by bivariate and multivariate analysis. There were 10 categories of risk factors listed in extended table (counted 19?). The mean influenza vaccination coverage in children aged 6 months to 14 years with risk in years 2018-2023 was 28.81% which was slightly higher than other studies conducted in the same country in previous seasons (23.9% in 2011/15, 15.6% in 2018/19 or 27.1% in 2009/10 coinciding with the influenza pandemic. Interestingly, the highest vaccination coverage (36.96%) during the study period was in 2020/21, 201 coinciding with the COVID-19 pandemic.

They observed that the variables age, number of risk factors and some of the risk factors such as asthma and diabetes were positively and statistically significantly associated with influenza vaccination in all seasons. Heart disease, coeliac disease, and obesity were negatively associated with influenza vaccination in virtually all seasons, such that patients with these three conditions were statistically significantly less vaccinated against influenza.  In all seasons, it was observed that the younger the children, the more they were vaccinated. Asthma, diabetes, haemoglobinopathies and clotting disorders were the risk factors most associated with vaccination. The findings are objectively and logically interpreted and explained. There objectives of the study and limitations are well outlined. Overall, this is very interesting data with objective, nonbiased explanations. Few corrections in editing might improve readability of the study such as second sentence of the abstract stating “there are risk factors that lead to vaccination” which needs to be more precise.

Comments on the Quality of English Language

It is well written paper with interesting data and concise discussion of the finding.

Author Response

The reply of the comments is in the attached file. 

REPLY REVIEWER 1

Comments 1: There were 10 categories of risk factors listed in extended table (counted 19?).

Response 1: Thank you for your comment. Table 1 shows de risk factors that Catalan Department of Health considered as indication for receiving the influenza vaccine in children. These risk factors were grouped into 10 categories to facilitate the statistical analysis.

The categories of risk factors are these:

-Asthma

- Heart diseases

- Coeliac disease

- Diabetis

- Haemoglobinopathies and clotting disorders

- Immunosuppression and neoplasms

- Neuromuscular, neurological disease, chromosomopathies and metabolopathies.

- Miscellaneous

- Nephropathies and hepatopathies

- Obesity.

Before, the review there was an extra category called “miopathy” that has been included in “Neuromuscular, neurological disease, chromosomopathies and metabolopathies”.

Comments 2: . Few corrections in editing might improve readability of the study such as second sentence of the abstract stating “there are risk factors that lead to vaccination” which needs to be more precise.

Response 2: Thank you for your comment. We have changed the sentence before for “Vaccination is recommended for certain diseases that could cause complications in case of flu infection”.

Reviewer 2 Report

Comments and Suggestions for Authors

Article describing vaccine coverage and the factors, including the COVID19 pandemic, that influenced childhood vaccination against influenza in Catalonia (2018-2023) in children with risk factors.

Overall, the article is well written and the conclusions are supported by the results.

I only have some minor comments.

Line 38. Please add the original reference to the WHO data and the year of estimates of that data.

Line 53. Add a reference for those prevalence data.

Lines 119-121. “The selection criteria for the variables for the multivariate analysis for each season were: that the data were statistically significant in the 120 bivariate analysis in the last season,…”

However, in Table 4 there are some variables (for example, country of origin) with a non-significant result in the bivariate analysis that were subjected to multivariate analysis (and some of them are significant...)

Material and methods. If possible, it would be enlightening for readers to know what the differences are between the rural and urban settings in this study. For example, is there any maximum number of people living in the village to consider it rural, or a type of main work activity?,...

Line 151. Table 3. Taking the 2018/19 season (without COVID) as reference, the authors could calculate the statistical significance of vaccination in the different seasons, to see in which of them the % vaccination is significant.

Table 4. It is very difficult to read in its current format. It is possible that using abbreviations for “bivariate analysis” and “multivariate analysis” can reduce the number of lines. Or consider including this Table 4 as supplementary material. Place the note “*Reference: not having a risk factor” as a footnote of the table. 

Lines 202-205. "This increase may be due to similarity...with risk factors." This argument is speculative since the authors do not show any data or reference on symptoms, fear or promotion of vaccination to support the statement.

Lines 223-226. “These differences can be explained by cultural reasons, due to the language barrier and the low educational level of the majority of immigrant parents (it has been observed that the rejection of the influenza vaccination is greater in parents with a high educational level [42]).

One would expect that, in fact, the lack of language knowledge would have made it more difficult for immigrants to follow the influenza vaccination recommendations given by health authorities for at-risk children.

Regarding education, in the present work the % of vaccinated children increases with a higher educational level: for example, in “parent education”, Primary Education has 295/1319 (22%) vaccinated children versus the University degree, which has 353/962 (37%). vaccinated.

Line 228-229. "The main reason for this 228 change may be fear of COVID-19." However, in the introduction, lines 68-69, the authors argue that “In addition, fear of SARS-CoV-2 infection and uncertainty led to the increased reluctance of some parents to go to health centres to vaccinate their children [18].”. These two messages seem contradictory. Does fear of COVID have a positive or negative effect on vaccination coverage? This message has to be clear since it is one of the conclusions of the work.

Lines 235-238. Although I agree with this line of argument, do the authors have any data on the number of children with hemoglobinopathies and bleeding disorders among their native and immigrant populations, to see if there is a significant difference and thus give more strength to this statement?

Lines 257-258. “As strong points, it should be noted that this is the first study of the effect of COVID19 on childhood influenza vaccination coverage in Spain.” Please, point out that the study was carried out only in children with risk factors; it is not a global population study.

Lines 259-261. I suggest moving the two lines to line 226.

Line 268. I suggest removing this sentence: "Further studies would need to be conducted to explore the reasons behind this." Is this actually a conclusion of the study?

Delete the word "Finally" on line 269.

Typographic. Write the entire text in the past tense (change “is” to “was”), lines 25, 30, 144,…

Round decimals as described in https://ec.europa.eu/eurostat/statistics-explained/index.php/Tutorial:Rounding_of_numbers#”

In the text, two significant digits (non-zero) are generally sufficient.

Author Response

The reply is in the attached file. 

REVIEWER 2

Comment 1: Line 38. Please add the original reference to the WHO data and the year of estimates of that data.

Response 1: Thank your for your comment. The original reference of the WHO is reference 1: “ World Health Organization. Global Influenza strategy 2019-2030. World Health Organization. Genveva: World Health Organitztion. 2019.[Consulted on 15 Mai 2023]. We have added this reference into the text.

Comment 2: Line 53. Add a reference for those prevalence data.

Response 2: Thank you for your comment. The reference is : “Barrio Cortes J, Suárez Fernández C, Bandeira de Oliveira M, Muñoz Lagos C, Beca Martínez MT, Lozano Hernández C, Del Cura González I. Enfermedades crónicas en población pediátrica: comorbilidades y uso de servicios en atención primaria. An Pediatr (Barce). 2020:93(3):183-193”. It is added as reference 8.

Comment 3: Lines 119-121. “The selection criteria for the variables for the multivariate analysis for each season were: that the data were statistically significant in the 120 bivariate analysis in the last season,…”

However, in Table 4 there are some variables (for example, country of origin) with a non-significant result in the bivariate analysis that were subjected to multivariate analysis (and some of them are significant...)

Response 3: Dear reviewer the variable “Country of origin” has only non-significant result in the bivariate analysis in the season 2020-2021 but we did the multivariate analysis because this data is very important. The season 2020-2021 was the first year of the COVID-19 pandemics and it was the only year with not significant result. So we can affirm that COVID-19 pandemics influenced the origin country of influenza vaccinated children. As we said in the “Statistical analysis” one of the selection criteria for the variables for the multivariate analysis was the clinical criteria of the researchers.

Comment 4: Material and methods. If possible, it would be enlightening for readers to know what the differences are between the rural and urban settings in this study. For example, is there any maximum number of people living in the village to consider it rural, or a type of main work activity?,...

Response 4: To classify the place of residence in rural or urban area we considered the number of people living in the village. A rural area is an area with less than 10,000 inhabitants. An urban area is an area with more than 10,000 inhabitants. We have added it in the line 100-102.

Comment 5: Line 151. Table 3. Taking the 2018/19 season (without COVID) as reference, the authors could calculate the statistical significance of vaccination in the different seasons, to see in which of them the % vaccination is significant.

Response 5: We can see the statistical significance with the confidence intervals seen in the IC 95% column. If the intervals do not overlap, it means that it is statically significant.

Comment 6: Table 4. It is very difficult to read in its current format. It is possible that using abbreviations for “bivariate analysis” and “multivariate analysis” can reduce the number of lines. Or consider including this Table 4 as supplementary material. Place the note “*Reference: not having a risk factor” as a footnote of the table.

Response 6: We have changed the table. We have removed some variables: gender and rurality because they were not significant. The rest of variables has been highlighted by color. The statistical significance has been highlighted with another color. We have abbreviated “bivariate analysis” and “multivariate analysis”.

Comment 7: Lines 202-205. "This increase may be due to similarity...with risk factors." This argument is speculative since the authors do not show any data or reference on symptoms, fear or promotion of vaccination to support the statement.

Response 7: This statement is speculative. We have added the observation “We assume that this increase …” to the text. (Line 216).

Comment 8: Lines 223-226. “These differences can be explained by cultural reasons, due to the language barrier and the low educational level of the majority of immigrant parents (it has been observed that the rejection of the influenza vaccination is greater in parents with a high educational level [42]).”

One would expect that, in fact, the lack of language knowledge would have made it more difficult for immigrants to follow the influenza vaccination recommendations given by health authorities for at-risk children.

Regarding education, in the present work the % of vaccinated children increases with a higher educational level: for example, in “parent education”, Primary Education has 295/1319 (22%) vaccinated children versus the University degree, which has 353/962 (37%). vaccinated.

Response 8: We are agree with your comment but in this case we mentioned the language barrier thinking in the fact that immigrant people do not understand which vaccine is administered. In the primary health center we observe that the majority of people accept the routine vaccines, but they do not accept influenza vaccine. In Spain, during 2018-2023 the influenza vaccine was not considered as a routine vaccine an it was less accepted. A barrier language can provide more influenza vaccination because immigrant people may do not understand which vaccine is administered.

Regarding education, in this work we can not differentiate between the education in immigrant people an in native population.

Comment 9: Line 228-229. "The main reason for this 228 change may be fear of COVID-19." However, in the introduction, lines 68-69, the authors argue that “In addition, fear of SARS-CoV-2 infection and uncertainty led to the increased reluctance of some parents to go to health centres to vaccinate their children [18].”. These two messages seem contradictory. Does fear of COVID have a positive or negative effect on vaccination coverage? This message has to be clear since it is one of the conclusions of the work.

Response 9: The lines 68-69 refers to routine childhood vaccination, not influenza vaccination. Influenza vaccination in Spain in 2018-2023 was not considered a routine childhood vaccine. We agree that the sentence can be contradictory. We has changed the sentence to : “In addition, fear of SARS-CoV-2 infection and uncertainty led to the increased reluctance of some parents to go to health centers to vaccinate their children of routine vaccines [19]”.

Comment 10: Lines 235-238. Although I agree with this line of argument, do the authors have any data on the number of children with hemoglobinopathies and bleeding disorders among their native and immigrant populations, to see if there is a significant difference and thus give more strength to this statement?

Response 10: We have some data about the number of hemoglobinopathies and bleeding disorders in Spain. But this data do not differenciate between native population and immigrant population. The majority of the hemoglobinopathies studied in this paper are “Sickle cell disease”. The Spanish registry of hemoglobinopathies referes that the 65% of the patients diagnosed with hemoglobinopathies were born in Spain, but de 51% of these were diagnosed in the neonatal screening. In the neonatal screening we can not differenciate between children of immigrant parents and children of native parents. The sickle cell disease is included in neonatal screening in Catalonia since 2015. For all these reasons we think that we can not report data to give more strength to this statement.

Comment 14: Typographic. Write the entire text in the past tense (change “is” to “was”), lines 25, 30, 144,…

Response 14: We have changed it.

Comment 15: Round decimals as described in https://ec.europa.eu/eurostat/statistics-explained/index.php/Tutorial:Rounding_of_numbers#”

In the text, two significant digits (non-zero) are generally sufficient.

Response 15: Thank your for your comment. We have rounded the significant digits in the text and we have put one decimal in the percentages.

Reviewer 3 Report

Comments and Suggestions for Authors

Well written manuscript! I appreciate the large data set and relatively clear description of the methodology. It is most definitely publishable after relatively minor revisions

- I would welcome if the data were available with the manuscript

- the result section looks like a one giant table. I like the details of the table but perhaps authors can keep the table and make a better explanation of the results in the text.

- there must be similar studies for different countries. In the discussion, there should be a comparison of this particular study to already published studies

Author Response

The reply of the comments is in the attached file. 

REVIEWER 3

Comment 1: I would welcome if the data were available with the manuscript.

Response 1:Thank You for your comment. We have changed the data availability statement for “The data that support the findings of this study are available from the corresponding author upon request”.

Comment 2: the result section looks like a one giant table. I like the details of the table but perhaps authors can keep the table and make a better explanation of the results in the text.

Response 2: We have changed the table. We have removed some variables: gender and rurality because they were not significant. The rest of variables has been highlighted by color. The statistical significance has been highlighted with another color.

Comment 3: There must be similar studies for different countries. In the discussion, there should be a comparison of this particular study to already published studies.

Response 3: We have added:

In line 222-229: “An Italian paper that studied the influenza coverage vaccination in children between 2010/11 and 2020/21 showed that in the season 2020/21 the coverage vaccination in children increased a 14,8% more than in previos seasons in the age group 2-4 years-old (coverage 19%), followed by the group 5-8 years-old and <2 years-old (increase +10%, coverage of 13,1% and +6,4%, coverage of 9,2% respectively) [28]. Another similar study conducted in South Korea showed that from 2019 to 2020 with the COVID-19 pandemic, the influenza vaccination coverage of children aged 12-18 increased from 27,8% to 43,5%[24].”

In line 233-239: “Fogel B et al conducted an investigation in Southcentral Pennsylvania to determine whether early season rates of influenza vaccination changed in a season when there was a concurrent COVID-19 pandemic. They observed that early vaccination rates were lower in 2020 (29.7%) compared with 2018 and 2019 (34.2% and 33.3% respectively) [21]. In the same line, Nogareda F et al described in 2023 the coverages of seasonal influenza vaccination in the Americas during 2019-2021. They reported that influenza vaccination coverage decresed a 9% for children [23].”
